# Is There a Need for Alcohol Policy to Mitigate Metal Contamination in Unrecorded Fruit Spirits?

**DOI:** 10.3390/ijerph17072452

**Published:** 2020-04-03

**Authors:** Dirk W. Lachenmeier

**Affiliations:** Chemisches und Veterinäruntersuchungsamt (CVUA) Karlsruhe, Weissenburger Strasse 3, 76187 Karlsruhe, Germany; lachenmeier@web.de; Tel.: +49-721-926-5434

**Keywords:** unrecorded alcohol, home-produced fruit spirits, metals, lead, cadmium, ethanol, health risk, risk assessment, margin of exposure

## Abstract

Unrecorded alcohol comprises all types of alcohol that is not registered in the jurisdiction where it is consumed. In some countries in Central and Eastern Europe as well as the Balkans, the majority of unrecorded alcohol consumption may derive from the home production of fruit spirits. Some studies found a high prevalence of lead and cadmium in such spirits. This article provides a quantitative comparative risk assessment using the margin of exposure (MOE) methodology for lead and cadmium, compared to ethanol, for unrecorded fruit spirits. For average concentration levels, the lowest MOE (0.8) was calculated for ethanol (alcohol itself). For lead, the MOE was 13 for moderate daily drinking and 0.9 for the worst-case scenario. For cadmium, the MOE was 1982 for moderate daily drinking and 113 for the worst-case scenario. The results of this study are consistent with previous comparative risk assessments stating that ethanol itself comprises by far the highest risk of all compounds in alcoholic beverages. Regarding metal contaminants, the risk of cadmium appears negligible; however, lead may pose an additional health risk in heavy drinking circumstances. Strategies to avoid metal contamination in the artisanal home production of spirits need to be developed.

## 1. Introduction

Unrecorded alcohol comprises all types of alcohol that is not registered in the jurisdiction where it is consumed [1,2,3]. This may be due to illegal or informal production, cross-border shopping, as well as the use of surrogate alcohol not originally intended for human consumption (e.g., automotive products, cosmetic or medicinal alcohol) [1,2,3,4]. In some countries in Central and Eastern Europe as well as the Balkans, the majority of unrecorded alcohol consumption may derive from home production of spirits from sugar-containing fruit materials such as cherries, plums, apples, pears or grapes, which grow in abundance in these countries [5,6,7,8,9,10,11]. The level of unrecorded consumption in the WHO European region varies between 3% (Austria) and 75% (Azerbaijan) (average 21%) of total alcohol consumption (calculated based on WHO data for 2016 [12]). Therefore, the health effects of unrecorded alcohol consumption comprise a substantial part of alcohol-related harm; however, this segment of alcohol consumption has been generally understudied [1,3,13,14]. An important topic of investigation is whether unrecorded alcohol would exhibit effects beyond recorded commercial alcoholic beverages. For example, it has been suggested that the differences in liver cirrhosis prevalence between countries that cannot be explained by the volume of consumption may be due to exacerbated alcohol-related effects due to unrecorded alcohol consumption. There are various hypotheses, such as that certain compounds in unrecorded alcohol may be causally related to its enhanced effects. Some studies point to so-called higher alcohols as culprits [10,11,15]. However, this hypothesis has not been universally accepted because many recorded fruit spirits contain similar levels of alcohol as their unrecorded counterparts [16,17,18].

So far, most chemical-toxicological studies on unrecorded alcohol in Europe were unable to detect levels of contaminants which may cause increased health effects including liver cirrhosis [1,19,20]. However, the studies typically detected higher levels of ethanol (i.e. alcoholic strength) in unrecorded alcohols [1,19,20], and epidemiological research provides evidence that unrecorded alcohols are typically consumed in more detrimental patterns of drinking [1,3]. Statistical research has shown that liver cirrhosis prevalence may be explained by differences in patterns of drinking alone [21]. One confounding factor may be the preference of unrecorded alcohol consumption by people of lower socioeconomic status [22]. However, based on the typical low sample sizes in previous chemical studies on unrecorded alcohol, the hypothesis that compounds other than ethanol may contribute to the health risks of unrecorded alcohol is worthy of investigation.

In 2019, two new studies on the chemical composition of unrecorded alcohol were published, namely, a pilot study on lead contamination of fruit spirits for own consumption from the Slovak Republic, but also including samples from Hungary [6], and a study on cadmium contamination in unrecorded plum spirits [7]. The studies found an extremely high prevalence of the two metals (lead 100%, n = 18; cadmium 97%, n = 35). The prevalence was much higher than in the current largest sample of European unrecorded alcohol from the Alcohol Measures for Public Health Research Alliance (AMPHORA) project (n = 115, prevalence lead: 53%; cadmium: 1%) [19,20]. However, the new studies [6,7] did not provide a quantitative risk assessment for the metals. Therefore, this article aims to provide such a risk assessment based on their data [6,7] using the margin of exposure (MOE) methodology [23,24].

## 2. Materials and Methods

The methodology for comparative risk assessment of compounds in alcoholic beverages using the margin of exposure method according to Lachenmeier et al. [25,26,27] was applied. Literature data about the occurrence of lead and cadmium in unrecorded alcohol according to Tatarková et al. [6,7] was used. For exposure scenarios, average and maximum concentrations of the compounds according to these studies [6,7] were applied. For comparison reasons, the scenarios for lead and cadmium in recorded and unrecorded alcoholic beverages according to Pflaum et al. [27] were included, as well as a risk assessment of ethanol itself according to the same authors. The toxicological endpoints used for dose-response modeling and the chosen points of departure for MOE assessment were strictly used according to Pflaum et al. [27]. The full methodology for comparative assessment is available in Lachenmeier et al. [25]. The MOE is defined as the ratio between the lower one-sided confidence limit of the benchmark dose (BMDL), or the no observed adverse effect level (NOAEL), and the estimated human intake of the same compound [20]. Regarding the interpretation of MOE values, if the value is lower, the risk is higher. For nongenotoxic substances, a 10–100-fold distance between the toxicological endpoint and human exposure would typically be demanded to exclude a health risk, meaning that the MOE should be 100 or higher if the assessment is based on animal data or 10 or higher if human data is used [20]. If the MOE is 1 or lower, this means that the human exposure reaches or exceeds the level which may cause adverse effects.

## 3. Results

Table 1 shows the corresponding MOEs for two different exposure scenarios of drinking one and four standard drinks per day combined with the average and maximum concentration. The first scenario comprising one drink per day (average concentration) could be treated as a moderate drinking scenario, while the last scenario (maximum concentration, four drinks per day) may be treated as a worst-case scenario. For average concentration levels, the lowest MOE (0.8) was calculated for ethanol (alcohol itself). For lead, the MOE was 13 for moderate daily drinking and 0.9 for the worst-case scenario. For cadmium, the MOE was 1982 for moderate daily drinking and 113 for the worst-case scenario. 

## 4. Discussion

The results of this study are consistent with previous comparative risk assessments [25,27,28,31] that ethanol itself comprises by far the highest risk of all compounds in alcoholic beverages. More refined calculations for MOE of ethanol using probabilistic methods (mean 1.3, CI 0.6–2.7) [26] also corroborate this result. This result holds true for recorded and unrecorded alcohol alike.

Regarding the metal contaminants, the results show that the risk of cadmium is negligible because an adequate safety margin is upheld for cadmium even in the worst-case scenario.

On the other hand, the previous result in a sample of unrecorded alcohol that the MOE of lead may be less than 10 [20] was confirmed in this study. As the risk assessment for lead is based on human data, a safety factor of 10 should be sufficient [20]. However, in worst-case scenarios for heavy drinkers, the MOE for lead falls below 1, which suggests a health risk. Interestingly, the same conclusion substantially holds true for recorded spirits, which, according to literature data, contain similar average amounts of lead, while only in worst-case scenarios the contamination level of unrecorded alcohol may be slightly higher.

As the MOE method is primarily intended for use by risk managers to set priorities [23,24], this study confirms the previous conclusion [27] that it should be a priority to reduce alcohol intake overall. Of course, this goal is rather difficult to achieve in the context of unrecorded, home-produced alcohol such as fruit spirits in Central European countries. Home-produced fruit spirits were generally found to have higher alcoholic strength (>40% vol) [6,7,19] than standard recorded spirits, which alone could lead to the ingestion of larger amounts of alcohol and (perhaps inadvertently) more detrimental drinking patterns.

Policy measures could include educating home distillers to measure, adjust and label ethanol content in their products, or implementing maximum alcoholic strengths for legal home production, which would naturally be difficult to enforce (see also [32] for policy options regarding unrecorded alcohol).

Policies must include mitigation measures for lead contamination as well. The MOE, at least in the worst-case scenario, appears to be much too low and outside typically accepted ranges. However, in the context of unrecorded fruit spirits, the difficulties are even greater than for ethanol, because lead and cadmium cannot be measured by producers, but instead, need to be determined using expensive laboratory trace analytical methods. Therefore, producers need to be educated about diligent production methods (i.e., avoidance of metal-leaching equipment unsuitable for food contact), or offered free analyses at state laboratories. 

## 5. Conclusions

The results of this study are in agreement with Tatarková et al. [6], i.e., that the issue of lead contamination is a relevant public health issue; however, the results disagree with their conclusion that cadmium could have health effects [7].

Alcoholic beverages, also including unrecorded alcohol, are multicomponent mixtures, which contain one or several toxic compounds besides ethanol [27]. Most of these compounds have different target organs and toxic mechanisms, and it is currently unclear if they may have additional or even synergistic effects. For example, lead targets the central nervous system and the kidney [29], while cadmium targets the lung, the kidney and the prostate [33].

The toxicological evaluation and risk assessment of combined exposures are currently only emerging, and, for example, the concept of combined or total margin of exposure (MOE_T_) assumes a common mechanism of action (see e.g. [31]). Due to the lack of knowledge about potential interactions between lead, cadmium, and ethanol, MOE_T_ currently cannot be calculated for unrecorded alcohol in order to evaluate the cumulative risk of ethanol, lead and cadmium. 

In line with previous research on home-produced spirits [34], prevalent metal contamination, specifically for lead, needs further research. To find strategies to avoid contamination, the sources need to be investigated (raw materials including drinking water used for dilution, pesticide treatment, distillation equipment, etc). Larger samples are also needed to refine the risk assessment. Within a framework of holistic alcohol control policies, measures to avoid metal contamination in unrecorded home-produced spirits should be included.

## Figures and Tables

**Table 1 ijerph-17-02452-t001:** The margin of exposure (MOE)^1^ of ethanol, lead and cadmium in unrecorded alcoholic beverages calculated for different drinking and contamination scenarios.

Agent	Scenario 1: One Standard Drink Per Day	Scenario 2: Heavy Drinker (Four Standard Drinks Per Day)
MOE for Average Concentration	MOE for Maximum Concentration	MOE for Average Concentration	MOE for Maximum Concentration (Worst Case)
Ethanol [25]	3	-	0.8	-
Lead, this study based on [6]	13	4	3	0.9
Lead, unrecorded alcohol in previous literature [25]	70	2	17	0.4
Lead, recorded spirits [25]	68	4	17	0.9
Cadmium, this study based on [7]	1982	453	496	113
Cadmium, unrecorded alcohol in previous literature [25]	∞^2^	349	∞^2^	87
Cadmium, recorded spirits [25]	2326	349	581	87

^1^ MOE = BMDL or NOAEL/Exposure. Ethanol: BMDL_10_ = 700 mg/kg bodyweight (bw)/day [28]; lead: BMDL_01_ = 0.0015 mg/kg bw/day [29]; cadmium: NOAEL = 0.01 mg/kg bw/day [30]. For details on the method and toxicological data, see [25,27]. Exposure data based on literature sources is referenced in column 1.^2^ The lemniscate symbol indicates that the MOE was not calculable as the average exposure was zero (i.e. below the detection limit of the applied analytical methodology).

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
