# Peer review of "Is There a Need for Alcohol Policy to Mitigate Metal Contamination in Unrecorded Fruit Spirits?"

_ijerph, 2020, doi:10.3390/ijerph17072452_

Round 1

Reviewer 1 Report

An excellent manuscript in its field, I suggest that the authors make a diagram as a circle showing which countries have the most contaminated water

Author Response

Thank you. Contaminated water may be only one source of the contamination of spirits. It is rather more likely that metal materials with food contact may be the source during illicit distillation. Nevertheless, I have added “drinking water” more specifically into the potential sources of contamination, which need to be investigated (line 148).

Regarding diagrams of water quality, which I feel may be out-of-scope, I would like to refer to the excellent reports of WHO Europe or the EU Commission:

https://www.eea.europa.eu/publications/state-of-water

Reviewer 2 Report

Abstract: for better understanding of the reader please consider amending the phrase "for the worst case" to "for the worst case scenario".

Methods: it shall be explained in methods that lead and cadmium contamination levels for producing the current risk assessment estimates were obtained from the previous published studies, and provide references for these studies. Otherwise the reader only understands this when reading Table 1 in the Results section.

Discussion: Probably the author shall consider discussing a bit contamination of unrecorded home produced spirits with phthalates, especially given that the author previously published papers on phthalates contamination. Otherwise the reader gets an impression that only the lead may pose additional risk beside ethanol in unrecorded fruits spirits. To mention that phthalates contamination is understudied due to small sample size of previously analyzed samples.

Author Response

Abstract: for better understanding of the reader please consider amending the phrase "for the worst case" to "for the worst case scenario".

Done.

Methods: it shall be explained in methods that lead and cadmium contamination levels for producing the current risk assessment estimates were obtained from the previous published studies, and provide references for these studies. Otherwise the reader only understands this when reading Table 1 in the Results section.

The methods section was clarified to more specifically point out that the occurrence data was from literature and not from own investigations. References to these studies were already included, but I have added them at a second instance to be absolutely clear.

Discussion: Probably the author shall consider discussing a bit contamination of unrecorded home produced spirits with phthalates, especially given that the author previously published papers on phthalates contamination. Otherwise the reader gets an impression that only the lead may pose additional risk beside ethanol in unrecorded fruits spirits. To mention that phthalates contamination is understudied due to small sample size of previously analyzed samples.

In all our previous investigations, we have seen phthalate levels in concentrations of being a health risk only in surrogate alcohols (e.g. cosmetic alcohol), which were denatured with diethyl phthalate when this practice was still allowed (Russia was one of the last countries finally prohibiting phthalates to denature alcohol). Additionally, none of the studies used in the materials section provided analytical data on phthalates. Some minor contamination of spirits with phthalates may occur by the use of plastic food contact materials (e.g. bags for fruit collection), but these residues were judged as not posing a health risk being below tolerable daily intakes and occur in recorded and unrecorded spirits alike (http://real.mtak.hu/35714/1/066.2016.45.1.17.pdf). For this reason, I am currently having a problem to find a suitable trajectory to include phthalates into the current discussion of metals and, therefore, decided rather not to touch the issue, which may be potentially confuse the reader why from all compounds, which might be potentially being toxic in unrecorded alcohol, the phthalates were pointed out in the discussion.

Reviewer 3 Report

The author examined a potential important topic—metal contamination in unregulated alcohol. 

(1) the nature of “unregulated alcohol” means numerous issues can be found, including contaminants. Why did the author only focus on the two metals, not any other “contaminants”? what was the rationale?

(2) The type of unregulated alcohol involved is rather limited. The author only re-analyzed data from three published reports (one is his own). The samples appear to be based in Europe. Unregulated alcohol is certainly not only seen in Europe, it is seen everywhere, especially in less developed regions.

(3) Page 1, line 33-35, the sentence "The levels of unrecorded consumption ...(average 21%) of total alcohol consumption" is this the percentage of total volume of alcohol consumed in Europe? ALso, it might be helpful to list the countries (regions) at the lower end ( 3%) and the high end (75%)

Author Response

The author examined a potential important topic—metal contamination in unregulated alcohol. 

(1) the nature of “unregulated alcohol” means numerous issues can be found, including contaminants. Why did the author only focus on the two metals, not any other “contaminants”? what was the rationale?

The rationale was that two new studies were published in 2019 regarding the health risk of lead and cadmium pointing our risks for public health. The prevalence of these metals was much higher than in our own previous investigations of unrecorded alcohol. Therefore, it became pertinent to conduct a quantitative risk assessment of these metals, which had not been conducted in the original studies in 2019. This rationale is stated in lines 56-65 of the introduction. I read through the rationale section and clarified some sentences.

(2) The type of unregulated alcohol involved is rather limited. The author only re-analyzed data from three published reports (one is his own). The samples appear to be based in Europe. Unregulated alcohol is certainly not only seen in Europe, it is seen everywhere, especially in less developed regions.

Actually this paper is intended for a special issue “Alcohol Control Policy and Health in Europe”. Therefore the aim was restricted to Europe. However, I am not aware of other recent chemical studies on these two metals in unrecorded alcohol from other countries. The segment of unrecorded fruit spirits is also more or less restricted to Europe, while in other parts of the world other sugar-containing matrices such as rice or grains are preferred for making homeproduced spirits.

(3) Page 1, line 33-35, the sentence "The levels of unrecorded consumption ...(average 21%) of total alcohol consumption" is this the percentage of total volume of alcohol consumed in Europe? ALso, it might be helpful to list the countries (regions) at the lower end ( 3%) and the high end (75%)

Yes, this is the percentage of volume of unrecorded alcohol consumption in relation to total alcohol consumption (recorded+unrecorded). This is data for the WHO European region, as has now been clarified. The low and high end countries were added as requested.

Round 2

Reviewer 3 Report

The author examined a potential important topic—metal contamination in unregulated alcohol.  The papers used the Margin of Exposure to calculate the potential risk of two metals and alcohol itself and he concluded that ethanol itself comprises by far the highest risk of all compounds in alcoholic beverages. However, my enthusiasm is very limited, due to the following concerns: (1) The analysis was merely based on three already published papers (references 6,7 and 25), no new data being introduced. (2) the nature of “unregulated alcohol” means numerous issues can be found, including contaminants. The author only focused on the two metals, not any other “contaminants” and he did not explain why; (3) The type of unregulated alcohol involved is rather limited. The author only re-analyzed data from three published reports (one is his own). The samples appear to be based in Europe. Unregulated alcohol is certainly not only seen in Europe, it is seen everywhere, especially in less developed regions.

Round 3

Reviewer 3 Report

The author has addressed all my concerns/comments, but I am still not enthusiastic about the significance of their findings.